# Characteristics of Varicella Breakthrough Cases in Jinhua City, 2016–2024

**DOI:** 10.3390/vaccines13080842

**Published:** 2025-08-07

**Authors:** Zhi-ping Du, Zhi-ping Long, Meng-an Chen, Wei Sheng, Yao He, Guang-ming Zhang, Xiao-hong Wu, Zhi-feng Pang

**Affiliations:** Department of Infectious Disease Prevention and Control, Jinhua Center for Disease Control and Prevention, Jinhua 321000, China; duzhiping1993@163.com (Z.-p.D.); longzhiping0403@163.com (Z.-p.L.); chenma416@163.com (M.-a.C.); swei0928@126.com (W.S.); yhe0917@163.com (Y.H.); zgmacn@126.com (G.-m.Z.)

**Keywords:** varicella, breakthrough cases, breakthrough intervals, dose intervals

## Abstract

**Background**: Varicella remains a prevalent vaccine-preventable disease, but breakthrough infections are increasingly reported. However, long-term, population-based studies investigating the temporal and demographic characteristics of breakthrough varicella remain limited. **Methods**: This retrospective study analyzed surveillance data from Jinhua City, China, from 2016 to 2024. Varicella case records were obtained from the China Information System for Disease Control and Prevention (CISDCP), while vaccination data were retrieved from the Zhejiang Provincial Immunization Program Information System (ISIS). Breakthrough cases were defined as infections occurring more than 42 days after administration of the varicella vaccine. Differences in breakthrough interval were analyzed across subgroups defined by dose, sex, age, population category, and vaccination type. A bivariate cubic regression model was used to assess the combined effect of initial vaccination age and dose interval on the breakthrough interval. **Results**: Among 28,778 reported varicella cases, 7373 (25.62%) were classified as breakthrough infections, with a significant upward trend over the 9-year period (*p* < 0.001). Most cases occurred in school-aged children, especially those aged 6–15 years. One-dose recipients consistently showed shorter breakthrough intervals than two-dose recipients (*M* = 62.10 vs. 120.10 months, *p* < 0.001). Breakthrough intervals also differed significantly by sex, age group, population category, and vaccination type (*p* < 0.05). Regression analysis revealed a negative correlation between the initial vaccination age, the dose interval, and the breakthrough interval (*R*^2^ = 0.964, *p* < 0.001), with earlier and closely spaced vaccinations associated with longer protection. **Conclusions**: The present study demonstrates that a two-dose varicella vaccination schedule, when initiated at an earlier age and administered with a shorter interval between doses, provides more robust and longer-lasting protection. These results offer strong support for incorporating varicella vaccination into China’s National Immunization Program to enhance vaccine coverage and reduce the public health burden associated with breakthrough infections.

## 1. Introduction

Varicella, commonly known as chickenpox, is an acute, highly contagious exanthematous disease caused by the varicella zoster virus (VZV). It is primarily transmitted through respiratory droplets and direct contact with vesicular fluid from infected individuals and is classified among the vaccine-preventable diseases with significant public health importance worldwide [1]. Varicella typically presents with fever, malaise, and a generalized vesicular rash. Although it is usually self-limiting, it may lead to serious complications such as bacterial superinfections, pneumonia, cerebellar ataxia, or encephalitis, particularly in adults and immunocompromised individuals [2].

In the absence of a universal vaccination program, varicella is endemic in most temperate climates, with over 90% of children acquiring infection before adolescence and the highest incidence and hospitalization rates observed in those under 10 years of age [3]. However, since the introduction of the live-attenuated varicella vaccine (VarV) in 1974, widespread immunization has significantly reduced the global incidence of varicella by over 90%, significantly reducing the associated disease burden [4]. Vaccination remains the most effective measure for varicella prevention and control, and many countries have incorporated the varicella vaccine into routine immunization schedules and adopted a two-dose immunization strategy to optimize protection and reduce the risk of infection [5].

Despite the proven effectiveness of VarV, breakthrough varicella—defined as varicella infection occurring more than 42 days after vaccination—has become increasingly common in regions with partial or incomplete implementation of two-dose regimens [6]. Studies have shown that vaccine-induced immunity may wane over time, especially in individuals who have received only a single dose [7,8]. Breakthrough infections are often characterized by milder and atypical clinical presentations, which may lead to underdiagnosis or delayed detection [9]. However, individuals with breakthrough varicella can still transmit the virus and contribute to outbreaks, especially in densely populated environments such as schools and childcare settings [10,11]. As a result, the growing incidence of breakthrough cases has raised concern over the long-term effectiveness of current immunization strategies and the need to re-evaluate dose timing and coverage levels. Previous research has mainly focused on the epidemiology of varicella and breakthrough cases, as well as the evaluation of vaccine effectiveness in different settings [12,13]. However, existing studies often emphasize general trends [14,15], with relatively few investigations into the dynamics of breakthrough infections, particularly in relation to the dose number, the timing of vaccination, and the interval between doses. In China, the varicella vaccine was introduced as a voluntary self-paid immunization, and it has not yet been included in the National Immunization Program (NIP), resulting in considerable variation in vaccination coverage across regions [16]. Although certain provinces recommend initiating varicella vaccination at one year of age and completing the two-dose regimen within the specified interval, catch-up vaccination and full implementation of the two-dose schedule are not uniformly practiced, potentially contributing to the rise in breakthrough cases [17]. At present, there remains a lack of long-term, population-based studies examining the detailed temporal and demographic patterns of breakthrough varicella under real-world vaccination conditions. Furthermore, few studies have quantitatively assessed the relationship between the age of initial vaccination, the dose interval, and the timing of breakthrough infections, which is particularly relevant for live-attenuated vaccines such as the VarV that rely on priming and boosting immune responses. Based on this, the present study analyzed varicella surveillance data from Jinhua City between 2016 and 2024, focusing specifically on the epidemiological characteristics and temporal distribution of breakthrough cases. Additionally, we evaluated the association between the initial vaccination age, the dose interval, and the breakthrough interval using statistical modeling. By identifying key risk factors and immunization patterns associated with breakthrough infections, our findings aim to inform evidence-based adjustments to vaccination strategies and provide a scientific basis for optimizing varicella prevention and control policies.

## 2. Data and Methods

### 2.1. Data Sources

Data on varicella cases were obtained from the China Information System for Disease Control and Prevention (CISDCP). Information on varicella vaccination was extracted from the Intelligent Service Information System of the Zhejiang Provincial Immunization Program (ISIS). It is important to note that varicella is not a notifiable disease in China, and thus, underreporting is likely. The reported cases may not fully represent the true disease burden, particularly for mild or atypical cases that do not require medical care. However, the CISDCP remains the most comprehensive source of routine infectious disease surveillance data available in China.

### 2.2. Case Extraction and Vaccination Data Matching

All varicella cases reported in Jinhua City between 2016 and 2024 were retrieved from the CISDCP using the current residential address and date of symptom onset as inclusion criteria. After the removal of duplicate and deleted records, vaccination information was matched through the ISIS using unique personal identification numbers. For cases lacking identification numbers, probabilistic record linkage was conducted based on a combination of name, sex, date of birth, and guardian’s name. A match was considered successful only if all four criteria were simultaneously satisfied. For successfully matched cases, the vaccination date and vaccine manufacturer information were extracted.

### 2.3. Definitions

Breakthrough case: A case of varicella occurring more than 42 days after administration of the varicella vaccine. Non-breakthrough case: A varicella case occurring within 42 days of vaccination. Breakthrough interval: The time interval between the most recent varicella vaccination and the onset of varicella symptoms. Unknown: Cases for which no vaccination record was identified in the ISIS. Unvaccinated case: A case in which the individual had not received any dose of the varicella vaccine. Age of initial vaccination: The age at which the first dose of the varicella vaccine was administered. Homologous vaccination: Two doses of varicella vaccine administered from the same manufacturer. Heterologous vaccination: Two doses of varicella vaccine administered from different manufacturers. Dose interval: The time interval between the first and second doses of the varicella vaccine.

### 2.4. Breakthrough Interval Calculation

Breakthrough cases were stratified into subgroups based on the dose number, sex, age, population category, and vaccination type. For each subgroup, the breakthrough interval was defined as the time elapsed (in months) between the administration of the most recent varicella vaccine dose and the onset of clinical symptoms. Summary statistics, including the minimum, maximum, median (*M*), 25th percentile (*P*_25_), and 75th percentile (*P*_75_), were calculated to describe the distribution of breakthrough intervals within each subgroup.

### 2.5. Heatmap Construction

Data were grouped according to the initial vaccination age (in months, *x*-axis) and the dose interval (in months, *y*-axis). For each grid cell defined by a unique combination of *x* and *y* intervals, the median breakthrough interval (*z* value) was calculated. A two-dimensional matrix of *z* values was then constructed, and color gradients were applied to visualize the duration of the breakthrough interval across the *x*–*y* plane.

### 2.6. Bivariate Cubic Regression Modeling

The initial vaccination age was defined as *x*, the dose interval as *y*, and the breakthrough interval as *z*. The model was specified as *z*~*k* + *x* + *y* + *x*^2^ + *xy* + *y*^2^ + *x*^3^ + *x*^2^*y* + *xy*^2^ + *y*^3^, where *k* is a constant. The model was fitted using the least squares method. Model performance was evaluated using the R-squared (*R*^2^) value and *p*-value, with *R*^2^ > 0.95 considered indicative of excellent model fit.

### 2.7. Statistical Analysis

All statistical graphs were created using GraphPad Prism8.0.1. Heatmap and surface plots were created using R4.2.0. Statistical analyses were performed using SPSS23.0. Categorical variables were described as frequencies and percentages [n (%)]. The Chi-square test or Chi-square test for trend was used to assess differences in proportions between groups. For non-normally distributed continuous data, the Wilcoxon rank-sum test (for two groups) or the Kruskal–Wallis rank test (for multiple groups) was used. All statistical tests were two-sided, and *p* < 0.05 was considered to be statistically significant.

## 3. Results

### 3.1. A Rising Trend in the Annual Proportion of Breakthrough Cases Was Observed in Jinhua City

From 2016 to 2024, a total of 28,778 varicella cases were reported in Jinhua City, among which 7373 (25.62%) were identified as breakthrough cases. Of these, 5444 (73.84%) were one-dose breakthrough cases and 1929 (26.16%) were two-dose breakthrough cases. The annual number of varicella cases exhibited an increasing trend before 2018, followed by a decline in subsequent years, with a slight rebound observed in 2024 (Figure 1A). Breakthrough cases were reported every year during the study period, and the proportion of breakthrough cases among all reported varicella cases showed a significant upward trend over time (χ^2^ for trend = 692.294, *p* < 0.001) (Figure 1B). Varicella cases were reported throughout the year, with two distinct seasonal peaks observed between May and July and between October and December. A total of 7816 cases (27.16%) were reported during May–July, and 10,990 cases (38.19%) during October–December (Figure 1C). Similarly, breakthrough varicella cases were reported in all months of the year, with two peak periods identified: April–June and October–December. Specifically, 1861 breakthrough cases (25.24%) occurred from April to June, and 3204 (43.46%) from October to December (Figure 1D).

### 3.2. Breakthrough Cases Were Primarily Observed Among School-Aged Children in Jinhua City

Among the 7373 breakthrough varicella cases reported, 4401 occurred in males (including 3253 one-dose cases and 1148 two-dose cases) and 2972 in females (including 2191 one-dose cases and 781 two-dose cases), yielding a male-to-female ratio of 1.48:1. The difference in the distribution of breakthrough cases between sexes was not statistically significant (*p* = 0.085). Breakthrough cases were mainly concentrated in four age groups: 1–5 years (740 cases), 6–10 years (2500 cases), 11–15 years (3286 cases), and 16–20 years (834 cases). Both one-dose and two-dose breakthrough cases were most commonly observed in the 6–10-year-old group, accounting for 29.13% and 14.95% of cases in that age group, respectively. With the exception of children under 1 year of age and adults over 26, the number of one-dose breakthrough cases consistently exceeded that of two-dose cases across all age groups, with statistically significant differences observed (*p* < 0.001). Regarding population categories, the majority of breakthrough cases occurred among primary school students (3779 cases) and secondary school students (2495 cases), followed by scattered children (510 cases) and children in nursery or daycare institutions (341 cases). In all population subgroups, one-dose breakthrough cases outnumbered two-dose cases, and these differences were also statistically significant (*p* < 0.001; Table 1).

### 3.3. Breakthrough Interval Was Influenced by Dose, Sex, Age, Population Category, and Vaccination Type

From 2016 to 2024, the breakthrough interval for one-dose varicella breakthrough cases in Jinhua City ranged from 1.80 to 193.10 months (*M* = 62.10, *P*_25_ = 37.55, *P*_75_ = 86.70). For two-dose breakthrough cases, the interval ranged from 1.50 to 254.80 months (*M* = 120.10, *P*_25_ = 87.90, *P*_75_ = 150.50). The difference in breakthrough intervals between the one-dose and two-dose groups was statistically significant (*p* < 0.001) (Figure 2A). Among males, the breakthrough interval ranged from 1.50 to 254.80 months (*M* = 105.90, *P*_25_ = 66.15, *P*_75_ = 142.70), while among females, it ranged from 1.50 to 233.40 months (*M* = 100.40, *P*_25_ = 62.55, *P*_75_ = 135.70). The difference between sexes was also statistically significant (*p* < 0.001) (Figure 2A). The median breakthrough intervals for the age groups of 1–5, 6–10, 11–15, and 16–20 years were 20.20 months (*P*_25_ = 11.65, *P*_75_ = 31.00), 79.50 months (*P*_25_ = 56.63, *P*_75_ = 97.70), 130.20 months (*P*_25_ = 105.60, *P*_75_ = 147.63), and 180.00 months (*P*_25_ = 162.23, *P*_75_ = 191.83), respectively. Differences among age groups were statistically significant (*p* < 0.001) (Figure 2B). The median breakthrough intervals for scattered children, kindergarten children, primary school students, secondary school students, college students, and other populations were 16.15 months (*P*_25_ = 8.60, *P*_75_ = 23.30), 38.70 months (*P*_25_ = 23.28, *P*_75_ = 55.55), 94.80 months (*P*_25_ = 73.30, *P*_75_ = 116.45), 151.00 months (*P*_25_ = 128.00, *P*_75_ = 169.40), 179.80 months (*P*_25_ = 85.10, *P*_75_ = 217.40), and 181.65 months (*P*_25_ = 165.63, *P*_75_ = 210.23), respectively. The differences in breakthrough intervals across these groups were statistically significant (*p* < 0.001) (Figure 2C). Regarding the vaccination type, the breakthrough interval for homologous vaccination ranged from 2.30 to 166.20 months (*M* = 55.80, *P*_25_ = 32.25, *P*_75_ = 80.75), while that for heterologous vaccination ranged from 1.80 to 193.10 months (*M* = 60.75, *P*_25_ = 37.00, *P*_75_ = 86.40). The difference between the two vaccination types was also statistically significant (*p* < 0.01) (Figure 2D).

### 3.4. Combined Effect of Initial Vaccination Age and Dose Interval on Breakthrough Interval

Among the two-dose breakthrough cases, 1620 (83.98%) individuals received their first dose between 12 and <24 months of age. The majority (1493 cases, 77.40%) received the second dose within 0–60 months after the first dose. Within this subgroup, individuals who received their first dose at 12–<24 months of age and had a dose interval of ≤60 months exhibited relatively longer breakthrough intervals. The longest breakthrough intervals were observed among those with a first dose at 12–<24 months and a dose interval of ≤15 months (Figure 3A). Further analysis using a bivariate cubic regression model revealed that both the initial vaccination age and the dose interval were negatively associated with the breakthrough interval (*R*^2^ = 0.964, *p* < 0.001). As the initial vaccination age increases, the correlation between the dose interval and the breakthrough interval becomes weaker. Likewise, as the dose interval increases, the influence of the initial vaccination age on the breakthrough interval diminishes. The joint effect of these two factors on prolonging the breakthrough interval is most evident when the initial vaccination age is within 36 months and the dose interval is within 60 months (Figure 3B).

## 4. Discussion

In recent years, China has made notable progress in the prevention and control of vaccine-preventable diseases, including varicella. Although the varicella vaccine was introduced in China in the late 1990s and gradually incorporated into various provincial immunization schedules, it remains a self-funded and non-mandatory vaccine. This status has resulted in substantial heterogeneity in vaccination coverage and implementation strategies across regions. To improve uptake, some provinces now recommend that children complete the two-dose varicella vaccination series as early as possible and have implemented policy incentives and community-based interventions to facilitate timely vaccine uptake. However, catch-up campaigns targeting older, unvaccinated children are rarely enforced, and adherence to the second dose remains suboptimal—particularly among migrant populations and those in less urbanized regions [18,19].

Our 9-year surveillance of varicella cases in Jinhua City from 2016 to 2024 revealed several important epidemiological trends. While the total number of reported cases increased steadily prior to 2018, a noticeable decline was observed following the implementation of the two-dose vaccination schedule, although a modest resurgence occurred in 2024. More notably, the proportion of breakthrough varicella cases exhibited a clear and statistically significant upward trajectory over the years, reflecting a shifting epidemiological landscape in the post-vaccine era and raising significant public health concerns. Similar patterns have been reported in other countries outside of China, highlighting that the increasing prevalence of breakthrough infections is a shared global challenge [20,21,22]. This trend raises critical questions regarding vaccine-induced immunity, the durability of protection, and the adequacy of current immunization strategies.

Breakthrough varicella typically occurs in individuals who have received at least one dose of the vaccine and often presents with milder clinical manifestations. However, these cases remain contagious and can still trigger outbreaks, especially in high-contact settings like schools and daycare facilities, due to parental unawareness, limited access, or a lack of reminder systems [23]. A key contributing factor to the rising incidence of breakthrough infections is the waning of vaccine-induced immunity over time, especially among those who have received only a single dose. Previous studies have demonstrated that seroconversion rates gradually decline 3 to 5 years post-vaccination, and cellular immune responses also diminish, increasing the risk of infection despite prior immunization [24]. In addition, incomplete or delayed administration of the second dose—often due to low parental awareness, limited access, or the absence of reminder systems—may result in suboptimal immune priming and inadequate booster responses, further compromising long-term protection [25]. Additionally, in school-aged populations, where close-contact networks are common, environmental factors such as inadequate ventilation, overcrowding, and suboptimal hygiene practices may further facilitate viral transmission—even within partially immunized cohorts—thereby increasing the risk of outbreak propagation despite vaccination efforts.

Seasonal trends in both varicella and breakthrough infections were clearly observed in our study. Total varicella cases exhibited two major peaks: one during May–July and another from October to December. Breakthrough infections displayed a similar seasonal pattern but peaked slightly earlier, particularly during April–June and again in the winter months. These trends coincide with the timing of school semesters in China and are likely driven by increased interpersonal contact in classroom settings, shared indoor facilities, and reduced outdoor activity during colder seasons [26,27]. This seasonal distribution aligns with the known epidemiological characteristics of varicella as a respiratory-transmitted infection, which tends to exhibit higher transmission rates in colder and more humid conditions [28,29]. The earlier onset of breakthrough peaks relative to overall varicella incidence may reflect increased vulnerability among vaccinated individuals, potentially due to waning immunity or inadequate secondary immune responses.

An analysis of demographic patterns revealed a predominance of breakthrough cases in males, although the sex difference was not statistically significant. Most cases occurred among children aged 6~15 years, particularly primary and secondary school students. This distribution aligns with previous research indicating that school-aged children serve as the predominant reservoir for breakthrough varicella infections, especially in regions lacking comprehensive two-dose immunization strategies, where partial vaccination coverage may be insufficient to prevent transmission in high-contact environments [30,31]. The elevated burden of cases in school environments may be attributed to frequent interpersonal interactions, crowded classrooms, and the challenge of early recognition due to atypical or milder symptoms often observed in breakthrough infections.

Interestingly, although the difference was not statistically significant, males in our cohort exhibited slightly longer breakthrough intervals compared to females. Similar trends have been reported in previous immunological research, suggesting that sex-based differences in immune responses may warrant further consideration in vaccine policy design [32]. Emerging evidence suggests that these differences may be modulated by hormonal and genetic influences and X-linked immune regulatory genes, which may affect immune memory and vaccine responsiveness [33,34]. Further studies are warranted to elucidate the underlying biological mechanisms and their implications for sex-specific vaccination strategies.

A notable strength of our study lies in the comprehensive analysis of breakthrough interval variations stratified by dose number, age, sex, population group, and vaccination type. The median breakthrough interval among one-dose recipients was significantly shorter than two-dose recipients, emphasizing the superior durability of immunity conferred by the two-dose regimen. When stratified by age, younger children exhibited shorter breakthrough intervals compared to adolescents and young adults, which may be attributed to their more recent vaccination and limited opportunities for natural immune boosting through environmental VZV exposure. Across population categories, college students and working adults demonstrated the longest breakthrough intervals, potentially reflecting the combination of earlier immune priming during adolescence and a longer time since their last vaccination. Notably, individuals who received heterologous vaccination—where different vaccine manufacturers were used for each dose—showed slightly longer breakthrough intervals compared to those who received homologous regimens. Although this difference was modest, it is consistent with previous evidence suggesting that mixed-manufacturer schedules may elicit comparable or even enhanced immunological memory responses [35].

Our analysis further investigated the joint impact of initial vaccination age and dose interval on the breakthrough interval, revealing a novel and clinically relevant interaction. The longest breakthrough intervals were observed among individuals who received their first dose between 12 and 24 months of age and completed the second dose within 15 months. However, as the initial vaccination age exceeded 36 months and the dose interval extended beyond 60 months, the strength of their association with the breakthrough interval progressively diminished. This pattern supports the immunological rationale for early immune priming and timely boosting to optimize memory B-cell development and antibody affinity maturation [36,37]. In the context of live-attenuated vaccines such as the VarV, initiating vaccination within the first two years of life promotes the generation of long-lived plasma cells and robust memory B-cell responses [38]. Conversely, delayed administration of the booster may fail to capitalize on this immunological window of opportunity, resulting in suboptimal immunogenicity and faster waning of vaccine-induced immunity.

Taken together, these findings underscore the importance of not only completing the two-dose schedule but also adhering to an appropriately timed interval to achieve maximum protective benefit. While our findings offer important insights into the real-world performance of varicella vaccination programs, several limitations must be acknowledged. First, this was a retrospective study based on passive surveillance data, and since varicella is not a nationally notifiable disease in China, underreporting may have occurred, especially for mild or subclinical cases. Second, vaccination records were linked through the immunization registry, and although matching was performed carefully, information bias cannot be completely ruled out. Third, our study lacked serological data, limiting the ability to directly assess immunogenicity. Lastly, we did not stratify breakthrough severity or viral genotypes, which would provide further evidence on the relationship between immune status and clinical outcomes. Future prospective studies incorporating serological assays, clinical severity assessments, and molecular characterization of circulating VZV strains are warranted to further elucidate the determinants of breakthrough varicella and to refine immunization strategies accordingly.

## 5. Conclusions

The present study highlights the growing burden of breakthrough varicella and the urgent need for evidence-based policy interventions. Enhancing two-dose vaccine coverage, promoting timely administration, and implementing catch-up programs for older children are critical steps in reducing breakthrough infections. Incorporating varicella vaccination into the National Immunization Program would help standardize practices and improve population-level immunity. Future prospective studies with immunologic follow-up are warranted to assess the durability of protection and to refine optimal dosing intervals for sustained disease control.

## Figures and Tables

**Figure 1 vaccines-13-00842-f001:**
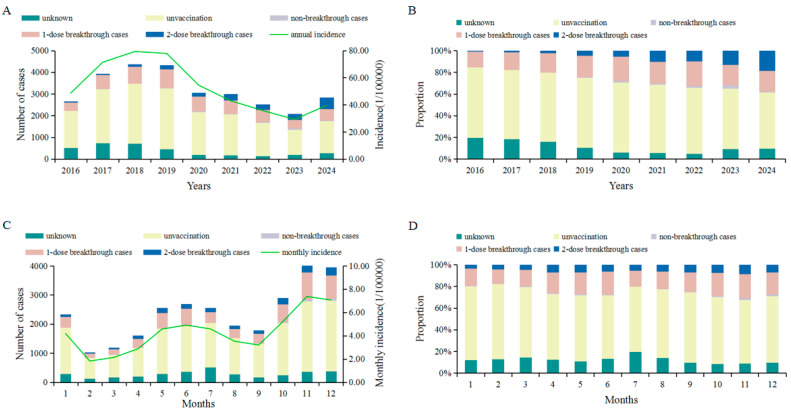
Epidemiological distribution of varicella and breakthrough cases in Jinhua City from 2016 to 2024. (**A**) Annual distribution of total varicella and breakthrough cases, along with annual incidence rate of varicella. (**B**) Annual proportion of breakthrough cases among all reported varicella cases. (**C**) Monthly distribution of reported varicella and breakthrough cases, and average monthly incidence rate of varicella. (**D**) Monthly proportion of breakthrough cases among total reported cases.

**Figure 2 vaccines-13-00842-f002:**
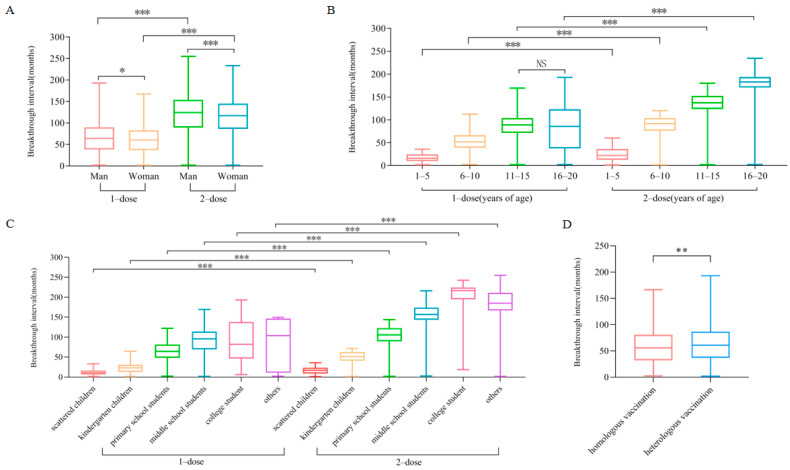
Group-wise comparison of breakthrough intervals. (**A**) Comparison of breakthrough intervals by sex. (**B**) Comparison of breakthrough intervals by age group. (**C**) Comparison of breakthrough intervals by population category. (**D**) Comparison of breakthrough intervals by vaccination type. NS: not statistically significant. * *p* < 0.05, ** *p* < 0.01, and *** *p* < 0.001 indicate statistically significant differences between groups.

**Figure 3 vaccines-13-00842-f003:**
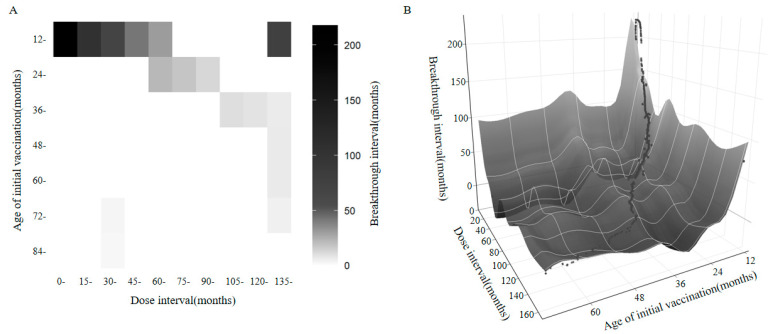
The effect of the initial vaccination age and the dose interval on the breakthrough interval. (**A**) A heatmap showing the combined effect of the initial vaccination age and the dose interval on the breakthrough interval. (**B**) A fitted surface plot based on a bivariate cubic regression model illustrating the relationship among the initial vaccination age, the dose interval, and the breakthrough interval.

**Table 1 vaccines-13-00842-t001:** Distribution characteristics of breakthrough cases in Jinhua City, 2016–2024 [n (%), % indicates the proportion within each subgroup].

Variables	Varicella Cases	Breakthrough Cases			*χ* ^2^	*p*
		1-Dose	2-Dose	Total		
Sex					0.034	0.085
Man	16,110	3253 (20.19)	1148 (7.13)	4401 (27.32)		
Woman	12,668	2191 (17.30)	781 (6.16)	2972 (23.46)		
Age (years)					189.090	<0.001
<1	735	0 (0.00)	0 (0.00)	0 (0.00)		
1~5	2061	535 (25.96)	205 (9.94)	740 (35.90)		
6~10	5671	1652 (29.13)	848 (14.95)	2500 (44.08)		
11~15	8739	2514 (28.77)	772 (8.83)	3286 (37.60)		
16~20	4750	730 (15.37)	104 (2.19)	834 (17.56)		
21~25	1707	13 (0.76)	0 (0.00)	13 (0.76)		
≥26	5115	0 (0.00)	0 (0.00)	0 (0.00)		
Demographic Groups					198.135	<0.001
Scattered children	1970	280 (14.21)	61 (3.10)	341 (17.31)		
Kindergarten children	1449	306 (21.12)	204 (14.08)	510 (35.20)		
Primary school students	8661	2653 (30.63)	1126 (13.00)	3779 (43.63)		
Middle school students	8842	2031 (22.97)	464 (5.25)	2495 (28.22)		
College student	2401	99 (4.12)	69 (2.87)	168 (6.99)		
Others	5455	75 (1.37)	5 (0.09)	80 (1.46)		

Percentages were calculated using the total number of varicella cases in each subgroup as the denominator.

## Data Availability

The original contributions presented in this study are included in the article. Further inquiries can be directed to the corresponding author(s).

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
