# Peer review of "Characteristics of Varicella Breakthrough Cases in Jinhua City, 2016–2024"

_vaccines, 2025, doi:10.3390/vaccines13080842_

Round 1

Reviewer 1 Report

Comments and Suggestions for Authors

Dear Authors,
Thank you for your worthwhile insight into a topic that is unfortunately always extremely relevant at worldwide levels. Reading your manuscript, I feel compelled to make some comments as below

  1. in the introduction section and here I cite: " In China, although the varicella vaccine is available, it has not been included in the national immunization program, resulting in considerable variation in vaccination coverage across regions[14]. Furthermore, catch-up vaccination and two-dose schedules are not uniformly implemented, potentially contributing to the rise in breakthrough cases[15]."
    It would be useful to briefly describe the recommended or most common vaccination schedule during the study period in order to make the data easier to interpret.
  2. In the result section, Table 1 needs to be fixed

  3. in the discussion section as here cited: "Further analysis using a bivariate cubic regression model demonstrated that both the age at first vaccination and the interval between doses were negatively associated with the breakthrough interval. In other words, earlier initiation of vaccination and shorter intervals between doses were associated with longer durations of protection. These findings may reflect the immunological advantage of early immune priming followed by timely boosting, which may help maintain vaccine-induced immunity before significant waning occurs."
    This is a crucial issue that requires further discussion. It needs to be explained as this could happen from an immunological standpoint, particularly for a live attenuated vaccine, when compared to existing immune response literature (prime-boost).
  4. It is necessary, in general, to boost the references section

Author Response

vaccines-3777932

Title: Characteristics of varicella breakthrough cases in Jinhua City, 2016-2024

Dear reviewer,

We sincerely appreciate your guidance, which has helped us improve the overall quality of the manuscript. The manuscript has now been revised (highlighted in red), and your questions are addressed in the point-by-point manner below.

Yours sincerely,

Dr Xiao-Hong Wu

Dr Zhi-Feng Pang

Comment 1

     In the introduction section and here I cite: " In China, although the varicella vaccine is available, it has not been included in the national immunization program, resulting in considerable variation in vaccination coverage across regions[14]. Furthermore, catch-up vaccination and two-dose schedules are not uniformly implemented, potentially contributing to the rise in breakthrough cases[15]."

It would be useful to briefly describe the recommended or most common vaccination schedule during the study period in order to make the data easier to interpret.

Response 1

Thank you for your valuable suggestion. We have revised the Introduction section accordingly to include a brief description of the recommended varicella vaccination schedule during the study period in China. This addition provides essential background for interpreting the data more accurately(line 84-91, highlighted in red).

Comment 2

In the result section, Table 1 needs to be fixed.

Response 2

Thank you for pointing this out. We have carefully revised and reformatted Table 1 to improve clarity and ensure that the data presentation is accurate and interpretable. The total column now clearly indicates the denominators used for calculating percentages, and the formatting has been fixed to align with journal standards.

Please see the revised Table 1 in the updated manuscript(highlighted in red).

Comment 3

In the discussion section as here cited: "Further analysis using a bivariate cubic regression model demonstrated that both the age at first vaccination and the interval between doses were negatively associated with the breakthrough interval. In other words, earlier initiation of vaccination and shorter intervals between doses were associated with longer durations of protection. These findings may reflect the immunological advantage of early immune priming followed by timely boosting, which may help maintain vaccine-induced immunity before significant waning occurs."
    This is a crucial issue that requires further discussion. It needs to be explained as this could happen from an immunological standpoint, particularly for a live attenuated vaccine, when compared to existing immune response literature (prime-boost).

Response 3

We agree that our previous discussion on the results was relatively superficial. In the revised manuscript, we have substantially expanded the discussion section, incorporating immunological perspectives to better explain the observed patterns(line 377-390, highlighted in red).

Comment 4

It is necessary, in general, to boost the references section.

Response 4

Following your suggestion, we have revised the manuscript and expanded the reference list. At present, the revised version includes 38 references, covering a broader range of recent studies, including international literature published within the last 5–10 years, to enhance the scientific depth and contextual relevance of our findings(line 511-643, highlighted in red).

Reviewer 2 Report

Comments and Suggestions for Authors

In this article, the authors retrospectively analyse the epidemiological trend of chickenpox in a province of China to support the thesis that the inclusion of paediatric vaccination in the national vaccination program could enhance vaccine coverage and reduce the public health burden associated with breakthrough infections. The article highlights the current lack of vaccination coverage in the province, with epidemics associated with school activities and a clear link to vaccination practices.

  • The main limitation of the study is that it is based on poor-quality data. Jinha is a city with a population of 7,050,683 (as of the 2020 census). Zhejiang consists of 90 counties with a population of 64.6 million. Therefore, the cases reported by the authors are completely unrealistic. This is not surprising, since the disease is not subject to mandatory reporting. However, the authors should warn readers of this situation in the description of the methods and not just mention the problem in the final section on limitations, as they have done.
  • Line 44. “Varicella is endemic worldwide, with children constituting the most susceptible population”. The authors' statement is correct, but only if there is no childhood vaccination programme in place. The authors could add that in temperate climates and in the absence of childhood varicella vaccination, more than 90% of people are infected by VZV and develop the disease before adolescence, with the highest incidence and hospitalization rates among children aged <10years [Otani, N.; Shima, M.; Yamamoto, T.; Okuno, T. Effect of Routine Varicella Immunization on the Epidemiology and Immunogenicity of Varicella and Shingles. Viruses 2022, 14, 588.
  • The statistical analyses are straightforward, their graphical presentation is effective, and the commentary is basic and essentially correct. The authors could improve the framing of their study within the literature by comparing their results with those of other countries. In its current version, the study has only 25 references, almost all of which are Chinese.

Author Response

vaccines-3777932

Title: Characteristics of varicella breakthrough cases in Jinhua City, 2016-2024

Dear reviewer,

We sincerely appreciate your guidance, which has helped us improve the overall quality of the manuscript. The manuscript has now been revised (highlighted in red), and your questions are addressed in the point-by-point manner below.

Yours sincerely,

Dr Xiao-Hong Wu

Dr Zhi-Feng Pang

Comment 1

The main limitation of the study is that it is based on poor-quality data. Jinha is a city with a population of 7,050,683 (as of the 2020 census). Zhejiang consists of 90 counties with a population of 64.6 million. Therefore, the cases reported by the authors are completely unrealistic. This is not surprising, since the disease is not subject to mandatory reporting. However, the authors should warn readers of this situation in the description of the methods and not just mention the problem in the final section on limitations, as they have done.

Response 1

Thank you for your insightful comment. Our data on varicella cases were obtained from the China Information System for Disease Control and Prevention(CISDCP), the CISDCP is the most comprehensive source of routine infectious disease surveillance data available in China. Although varicella is not a nationally notifiable disease in China and underreporting may therefore occur, our analysis was based on nine years of continuous surveillance data. As part of the revision, we have provided additional details in the “Data Sources” section of the Methods to enhance transparency and help readers gain a clearer understanding of the data utilized in this research(line 115-119, highlighted in red).

Comment 2

Line 44. “Varicella is endemic worldwide, with children constituting the most susceptible population”. The authors' statement is correct, but only if there is no childhood vaccination programme in place. The authors could add that in temperate climates and in the absence of childhood varicella vaccination, more than 90% of people are infected by VZV and develop the disease before adolescence, with the highest incidence and hospitalization rates among children aged <10years [Otani, N.; Shima, M.; Yamamoto, T.; Okuno, T. Effect of Routine Varicella Immunization on the Epidemiology and Immunogenicity of Varicella and Shingles. Viruses 2022, 14, 588.

Response 2

We have revised the Introduction section and cited relevant references(line 55-57, highlighted in red).

Comment 3

The statistical analyses are straightforward, their graphical presentation is effective, and the commentary is basic and essentially correct. The authors could improve the framing of their study within the literature by comparing their results with those of other countries. In its current version, the study has only 25 references, almost all of which are Chinese.

Response 3

Following your suggestion, we have revised the manuscript and expanded the reference list. At present, the revised version includes 38 references, covering a broader range of recent studies, including international literature published within the last 5–10 years, to enhance the scientific depth and contextual relevance of our findings(line 511-643, highlighted in red).

Reviewer 3 Report

Comments and Suggestions for Authors

Dear Authors,
The paper is very interesting because it deals with a preventable disease, and therefore requires close vigilance.
I would like to make a few comments.
In the Statistical Analysis section, it states that ANOVA or Kruskal-Wallis was used depending on the normality.
I would like to know which statistical test was used to determine the normality of the variables and the results of these tests, as well as which variables ANOVA was used, and which variables used nonparametric tests.
Furthermore, in health sciences, variables do not usually have a normal distribution due to the actions of health professionals.
I recommend always using nonparametric tests.
In Table 1, the total column is misleading because it is not clear what the denominator is for calculating percentages.
On line 180, is the interval 1.50 - 254.80 months correct?
The graphs were not created with SPSS. I assume they were created with R. They should include the packages used and the tests used for the comparisons. If the comparison was made with another program (GraphPad Prism?), this should be indicated.
Cubic regression should be in the Statistical Analysis section, even if it was performed post hoc.

Author Response

vaccines-3777932

Title: Characteristics of varicella breakthrough cases in Jinhua City, 2016-2024

Dear reviewer,

We sincerely appreciate your guidance, which has helped us improve the overall quality of the manuscript. The manuscript has now been revised (highlighted in red), and your questions are addressed in the point-by-point manner below.

Yours sincerely,

Dr Xiao-Hong Wu

Dr Zhi-Feng Pang

Comment 1

In the Statistical Analysis section, it states that ANOVA or Kruskal-Wallis was used depending on the normality.
    I would like to know which statistical test was used to determine the normality of the variables and the results of these tests, as well as which variables ANOVA was used, and which variables used nonparametric tests.
    Furthermore, in health sciences, variables do not usually have a normal distribution due to the actions of health professionals.
    I recommend always using nonparametric tests.
Response 1

Thank you for your insightful comment. Upon re-evaluating the manuscript, we noticed inconsistencies in the reporting of statistical methods. We have revised the relevant sections accordingly to ensure accuracy and clarity(line 175-184, highlighted in red).

Comment 2

In Table 1, the total column is misleading because it is not clear what the denominator is for calculating percentages.

Response 2

We have carefully revised and reformatted Table 1 to improve clarity and ensure that the data presentation is accurate and interpretable. The total column now clearly indicates the denominators used for calculating percentages, and the formatting has been fixed to align with journal standards.

Please see the revised Table 1 in the updated manuscript(highlighted in red).

Comment 3

On line 180, is the interval 1.50 - 254.80 months correct?

Response 3

The originally reported breakthrough interval of 1.50–254.80 months in Line 180 was correct. However, upon further review, we discovered an error in the reporting of the interval range for two-dose breakthrough cases, which has now been corrected. As shown in the revised Figure 2A and Figure 2C, the value of 254.80 months corresponds specifically to male two-dose recipients and individuals in the 'other' population group. We have updated the text accordingly to ensure consistency and accuracy(line 236, highlighted in red).

Comment 4

The graphs were not created with SPSS. I assume they were created with R. They should include the packages used and the tests used for the comparisons. If the comparison was made with another program (GraphPad Prism?), this should be indicated.

Response 4

We have revised the Methods section to provide a clearer and more detailed description, ensuring better transparency and reproducibility of the study(line 110-184, highlighted in red).

Comment 5

Cubic regression should be in the Statistical Analysis section, even if it was performed post hoc.

Response 5

 We have made the corresponding modifications in the revised manuscript(line 163-184, highlighted in red).

Round 2

Reviewer 2 Report

Comments and Suggestions for Authors

the authors revised the manuscript

Reviewer 3 Report

Comments and Suggestions for Authors

Dear Authors,
Thank you very much for answering my questions and clarifying all my comments.
Great work.